# Energy-Efficient Adaptive Sensing Scheduling in Wireless Sensor Networks Using Fibonacci Tree Optimization Algorithm

**DOI:** 10.3390/s21155002

**Published:** 2021-07-23

**Authors:** Liangshun Wu, Hengjin Cai

**Affiliations:** School of Computer Science, Wuhan University, Wuhan 430072, China; wuliangshun@whu.edu.cn

**Keywords:** sensing, energy-saving, duty cycles, Fibonacci tree optimization

## Abstract

Wireless sensor networks are appealing, largely because they do not need wired infrastructure, but it is precisely this feature that renders them energy-constrained. The duty cycle scheduling is perceived as a contributor to the energy efficiency of sensing. This paper developed a novel paradigm for modeling wireless sensor networks; in this context, an adaptive sensing scheduling strategy is proposed depending on event occurrence behavior, and the scheduling problem is framed as an optimization problem. The optimization objectives include reducing energy depletion and optimizing detection accuracy. We determine the explicit form of the objective function by numerical fitting and found that the objective function aggregated by the fitting functions is a bivariate multimodal function that favors the Fibonacci tree optimization algorithm. Then, with the optimal parameters optimized by the Fibonacci tree optimization algorithm, the scheduling scheme can be easily deployed, and it behaves consistently in the coming hours. The proposed “Fibonacci Tree Optimization Strategy” (“FTOS”) outperforms lightweight deployment-aware scheduling (LDAS), balanced-energy scheduling (BS), distributed self-spreading algorithm (DSS) and probing environment and collaborating adaptive sleeping (PECAS) in achieving the aforementioned scheduling objectives. The Fibonacci tree optimization algorithm has attained a better optimistic effect than the artificial bee colony (ABC) algorithm, differential evolution (DE) algorithm, genetic algorithm (GA) algorithm, particle swarm optimization (PSO) algorithm, and comprehensive learning particle swarm optimization (CLPSO) algorithm in multiple runs.

## 1. Introduction

Wireless Sensor Network (WSN) is a network of thousands of low-cost miniature devices capable of processing, communicating wirelessly, and sensing, which runs on a limited battery. Since we typically expect WSN sensors to last from several months to one year without recharging, energy efficiency is essential. Much previous work has referred to energy-efficient communication (media access control and routing) in WSN ([1,2,3,4], for example); energy-efficient sensing, however, has received little attention. Results of recent measurements indicate that sensing uses comparable amounts of energy as wireless communication [5]. Additionally, sensing frequency is greater than communication frequency—for example, a sensor that monitors residential fire in forests only triggers an alarm when temperatures exceed a threshold. Therefore, it is prime time we called for efforts to study energy-saving strategies of sensing.

Dynamic events are not easily captured as they come and go and can only be observed by continuous monitoring. Energy efficiency is a serious concern if sensors are always ON. Programming sensors’ work/sleep cycles (or duty cycles) is seen as a helpful response measure. When the node is in SLEEP mode, only the low-power timer remains active should it be necessary to wake it up. Hence, the energy consumed during the sleep duration is only a tiny fraction of that in the working duration.

While many studies have investigated sensing scheduling problems, they differ in their design assumptions and objectives because of the multifarious business needs of applications. Based on the belief of building up mathematical foundations and establishing synergy between detection accuracy and energy depletion (as the central theme of this paper), we suggest that the main contributions of this paper are:A novel system model of wireless sensor network is introduced;An adaptive sensing scheduling strategy based on event occurrence behavior is proposed;The scheduling problem is formulated as an optimization problem and is solved by a Fibonacci tree optimization algorithm.

The rest of the paper proceeds as follows: Section 2 reviews related work. Section 3 presents the system model. Section 4 introduces the Fibonacci tree optimization algorithm. Section 5 explains why the Fibonacci tree optimization algorithm is selected for parameter optimization and examines the advantages of the proposed FTOS strategy. Section 6 implements experiments. Section 7 discusses the deficiencies and unresolved issues. Finally, Section 8 draws the conclusion.

## 2. Related Work

### 2.1. Sensing Scheduling Strategies

The research on energy-saving sensing scheduling strategies involves two levels. The first level is adaptive duty cycle scheduling, concerning active time, nap time and the idle listening period. The goal is to best model the event arrival patterns. The second level, collaborative sensing, considers the sensing coverage: how numerous sensor nodes (not just a single node) with different spatiotemporal coordinates in a public area can cooperate to achieve adequate coverage, which is conducive to global optimization of energy consumption.

We know that sensors consume the most energy when they are on duty and the least when they are sleeping. As a result, almost all scheduling strategies, such as ELECTION [6] and additive increase/multiplicative decrease (AIDM) [7], make full use of the energy-saving characteristic of the sleep mode. DANCE [8] improves AIDM [7] by incorporating the behavior of neighboring nodes into consideration scope. In DANCE, the sensor abandons its task if its neighbor has already done it, thus down-scaling the energy inefficiency. The authors argue that the data sampling rate affects the computing and communication load on the central server. Controlling the sampling rate within a specific range (if it exceeds, then the central server and nodes renegotiate a range) through Kalman filtering is believed to be capable of tackling this problem [9].

Jothiraj et al. [10] recommend fine-tuning the sensing frequency of each sensor dynamically. As the sensory readings increase, the detection accuracy improves. The challenge, however, lies in modeling the actual world. Maintaining the greatest possible coverage is a primary goal when designing a sensor network. Chen et al. [11] defined a sensing coverage metric that measures a wireless sensor network’s QoS (quality of service). Furthermore, a polynomial-time complexity optimization algorithm using graphing theory and computational geometry is proposed to achieve optimum coverage [12] broadens the work in [11] by enhancing the algorithm. The first work that considers both energy consumption and sensing coverage was completed by [13], which introduces an energy-efficient surveillance system (ESS). This was followed by lightweight deployment-aware scheduling (LDAS) [14,15], probing environment and adaptive sensing (PEAS) [16], probing environment and collaborating adaptive sleeping (PECAS) [17], and randomized independent scheduling (RIS) [18], and so forth. These studies make the following two common assumptions: (1) each sensor is power-constrained, and (2) the network is expected to run for a long time. Other assumptions include:Network Structure. The network structure can be flat or hierarchical.Sensor Placement. The sensing coverage is usually affected by how sensors are initially placed. In most cases, the sensors follow a random distribution or two-dimensional (2-D) Poisson distribution [12,19].Sensing Area: The sensing area can be 2-D circular or 3-D spherical.Time Synchronization. The sensors are synchronized in time, and can be woken up simultaneously for the next scheduling round.Failure Model. Almost all the studies assume that sensors fail when energy is exhausted.Sensor Mobility. Most studies have assumed that the sensor is immobile.Location Information. These studies typically associate location information with whether, or how much, a sensor’s sensing area overlaps with its neighbor’s.Distance Information. The distance information can be inferred from the location information.

Randomized independent scheduling (RIS) [18] can extend the sensor lifetime and obtain an asymptotic *k*-coverage, and, it is simple. RIS does not need to provide location information or distance information, nor does it need adjustable transmission range and mobility. However, it is based on strict distribution assumptions—for example, the sensor is Poisson/uniform/grid distributed, the sensing range follows a uniform distribution and the network is flat and two-dimensional.

Lightweight deployment-aware scheduling (LDAS) [15], probing environment and adaptive sensing (PEAS) [16], probing environment and collaborating adaptive sleeping (PECAS) [17], and so forth, make relatively loose assumptions. LDAS assumes that the sensor nodes do not own a positioning device, such as GPS, thus forces the sensors achieving the desired coverage by static sensing. PEAS consists of two mechanisms: sensing and adaptive sleep scheduling, and requires high sensor density. PECAS is an updated version of the PEAS. It posts its remaining/available hours in response to neighbors to avoid misperceptions. This leads to an increase in communication energy depletion. Balanced-energy scheduling (BS) [20] is a balanced-energy scheduling model designed for dense sensor networks. It distributes the sensing and communication tasks to all sensor nodes in the cluster. The assumptions made by the reviewed studies are listed in Table 1.

Applications differ in their necessities. Hence, the served sensor networks have diverse design objectives and priorities. We summarize these design objectives as follows:Maximizing Network Lifetime. This is a goal that is hard not to consider.Sensing Coverage. A network is said to achieve *k*-coverage if any event occurs within the jurisdiction of at least *k* sensors. 1-overage is a minimum requirement for WSN.Network Connectivity. The developers applaud a model that offers a particular network connectivity needed by the application, but this requires a very high sensor density.Balanced Energy Usage. In the case of a sensing coverage breach, when a node runs out of power before others, some studies endeavor to spread the energy utilization evenly among each node.Simplicity. Sensors have exceptionally restricted memory space and limited computation power. For this reason, simple schemes are more popular.Robustness: Robustness measures how well a network can withstand downtime and crashes.

Adaptive self-configuring sEnsor networks topologies (ASCENT) [18], PEAS, PECAS, Low-energy adaptive clustering hierarchy (LEACH-GA) [23], IBLEACH [24], energy-efficient surveillance system (ESS) [25] and BS [20] do not take complete coverage of the region as their primary goal. Cooperative spectrum sensing (CSS) [22], however, is the opposite. The coverage configuration protocol (CCP) [21] introduced the concept of interconnected sensor coverage. It devised an approximation algorithm to construct a topology with near-optimal sensor coverage. The central controller periodically selects sensors along a trajectory until the target area is fully covered.

Most schemes strive to attain an energy balance [28], for example, the distributed self-spreading algorithm (DSS) [26] and intelligent deployment and clustering algorithm (IDCA) [27]. In the DSS, the sensor nodes are initially deployed randomly and move because of the influence exerted by nearby nodes. In the IDCA, however, the remaining energy level of the node determines whether it moves or not. The idea behind DSS and IDCA is to reduce the residual capacity differential between nodes.

Many studies envisage network connectivity with sensing coverage, for example, [29,30,31]. When the transmission range of the sensor node is at least twice its sensing range, *k*-coverage leads to *k*-connectivity [29,30]. Typically, high connectivity ensures high robustness, but one result of high connectivity is that data conflicts between nodes can seriously affect data transfer rates. The results of recent research presented in [32] are not based on the assumption that the transmit range of the sensor nodes is not less than twice their sensing range. Dhumal et al. [31] considered a tiered sensor network that includes sensors that could fail and they discussed the sensing coverage, network connectivity, and network diameter. In [19], the authors proposed an optimal deployment strategy to achieve two connections that fully cover all communication and sensing ranges.

The design objectives of the reviewed studies are summarized in Table 2.

### 2.2. Optimization Algorithms for Multi-Modal Function

Population or single solution search-based optimization algorithms (viz., meta, hyper-heuristic) have solid global search ability. Typical examples include inter alia evolutionary and swarm intelligence algorithms, such as the genetic algorithm (GA) [33], differential evolution (DE) algorithm [34], particle swarm optimization (PSO) algorithm [35], comprehensive learning particle swarm optimization (CLPSO) [36] and artificial bee colony (ABC) algorithm [37]. These algorithms are essentially methods by trial and error. It can take many thousands or even millions of iterations to converge. New emerging algorithms, such as the multi-layered gravitational search algorithm (MLGSA) [38] and quantum tabu search (QTS) algorithm [39], improve classical versions by avoiding premature convergence or being trapped in local optima. When addressing practical issues, we search for the global extremum of a complicated or unknown function, but just finding one local minimum of a relatively simple but very high-dimensional function can also be a formidable challenge; for example, the Multi-modal Function Optimization (MFO) [40]. Given a multi-modal problem, the optimization task is to identify the greatest number of optimal solutions (global and local) to help the decision-maker better understand the problem at hand.

It has developed many techniques to locate local optimums. These techniques are termed “niching” methods. A niching method can be built into a standard search-based optimization algorithm to identify several optimal or sub-optimal solutions sequentially or concurrently. Sequential approaches discover optimal solutions gradually over time, while concurrent methods encourage and maintain many stable sub-populations within a single population. Conventional niching techniques include crowding, fitness sharing, decommissioning, covariance matrix adaptation, cleaning, species conserving, and so forth. More recently, some variants of meta-heuristic algorithms, such as an ant colony system with nonlinear pheromone update (ACS-NP) [41], and comprehensive learning particle swarm optimization with local search (CLPSO-LS) [42], have incorporated niching methods. Despite niching methods’ first appearance over 30 years ago (in the 1980s), niching methods (or multi-modal optimization) are being resurrected as an increasingly important research subject, attracting researchers from a broad spectrum of research areas, including Evolutionary Computation (EC) [43] and Swarm Intelligence (SI) [42].

The Fibonacci tree optimization algorithm (FTO) [44] is a sophisticated optimization algorithm. It resolves the optimal solution of the problem by alternating iterations of global scanning and local scanning. It fully utilizes computer memory to save the optimization process. FTO can provide a reasonable approximation of a global optimum for a function with ample search space. In each iteration, the golden ratio separation is used to compress the search space, so the local optimal solution can also be achieved. It particularly applies to multi-peak/multi-modal function optimization.

The comparison of features of the aforementioned optimization algorithms is shown in Table 3.

## 3. System Model

Table 4 lists the essential notations used in the system model.

### 3.1. Basic Assumptions

The model operates on a 2-D map with many devices, each equipped with sensors to perform specific tasks. We make the following basic assumptions:**A1**: The network structure is flat. All nodes are homogeneous with the same energy budget.**A2**: The positions of sensor nodes are uniformly distributed.**A3**: The sensing area is 2-D.**A4**: The transmission/sensing radii are tunable.**A5**: Time is asynchronous. It is hard to coordinate sensors without a central controller but, otherwise, the central controller incurs a performance penalty. So, sensors embrace asynchronous scheduling—every sensor autonomously decides its duty cycle without synchronizing with each other.**A6**: The sensor node can be movable.**A7**: Location information is unknowable.**A8**: Distance information is unknowable.

### 3.2. Location Distribution and Sensing Radii

We assume the participants move around in *M* randomly. The locations of all *N* participants follow a uniform distribution. To accommodate different sensing radii, the sensor can adjust its transmission range *r* from $r_{min}$ to $r_{max}$, which is usually accomplished by varying the transmitting power (see Figure 1).

### 3.3. Event Occurrence and Detection

We assume the events occur randomly and independently in *M*, subject to a Poisson process of Poisson rate λ. The Poisson process is a stochastic process used to model random and independent events. Furthermore, it is assumed that the measurement is efficient, which means all the events occurring within the sensing range will be expediently detected. Therefore, the probability that *q* events processed by a sensor with sensing radius *r* in duration *t* is
(1)$$\mathrm{Pr}\left(C(t)=q\right)\;=\frac{\exp\left\{-\lambda^{\prime }\pi r^{2}t\right\}{\left(\lambda^{\prime }\pi r^{2}t\right)}^{q}}{q!}=\frac{\exp\left\{-\lambda t\right\}{\left(\lambda t\right)}^{q}}{q!},$$
where C(t) is the counting process of events and $\lambda =\lambda^{\prime }\pi r^{2}$ represents the average Poisson rate in the circular sensing region of the sensor. By leveraging the properties of Poisson’s process, we have:(2)$$E\left[C(t)\right]\;=\hat{q}{|}_{\left[0,t\right]}=\lambda t,$$
where $E\left[\cdot \right]$ is the expectation operator and $\hat{q}{|}_{\left[0,t\right]}$ denotes the expected number of events processed in [0,t].

An event at a distance of *d* from the sensor has a probability of being detected:(3)$$\mathrm{Pr}\left(d\right)=\left\{\begin{array}{c} \begin{array}{cc} 1, & \mathrm{if}\;d<r_{\min}\\ \exp\left\{-\xi_{1}{\left(d-r_{min}\right)}^{\xi_{2}}\right\}, & \mathrm{if}\;r_{\min}<d<r_{\max}\\ 0, & \mathrm{if}\;d>r_{\max} \end{array} \end{array}\right.,$$
where $\xi_{1}$ and $\xi_{2}$ are parameters determined by the sensor physical characteristics, which reflects the probability attenuation coefficients with distance.

In sleep time, all events are considered missed. Let $\Gamma_{S}$ be the average number of missed events, then
(4)$$\Gamma_{S}=E\left[C(S)\right]=\lambda S.$$

During working hours, the average number of missed events, $\Gamma_{W}$, can be calculated as
(5)$$\Gamma_{W}=E\left[C(W)\right]\left(1-\mathrm{Pr}\left(d\right)\right)=\lambda W\left(1-\mathrm{Pr}\left(d\right)\right).$$

The event space is
(6)$$\Gamma =E\left[C\left(W+S\right)\right]=\lambda \left(W+S\right).$$

Now we define the detection accuracy μ as a quantifiable indicator of event missing
(7)$$\mu =1-\frac{\Gamma_{S}+\Gamma_{W}}{\Gamma }.$$

It follows from (Equation 3)–(Equation 6) that
(8)$$\mu =\left\{\begin{array}{c} \begin{array}{cc} \frac{W}{S+W}, & \mathrm{if}\;d<r_{\min}\\ \frac{W}{S+W}\exp\left\{-\xi_{1}{\left(d-r_{min}\right)}^{\xi_{2}}\right\}, & \mathrm{if}\;r_{\min}<d<r_{\max}\\ 0, & \mathrm{if}\;d>r_{\max} \end{array} \end{array}\right..$$

### 3.4. Energy Depletion

We consider the sensing energy model introduced in Lee et al. [8]:(9)$$e_{W}=\gamma {\left(d-r_{min}\right)}^{\alpha }+\varepsilon_{W},$$
where γ,α are sensor-specific constants; in exemplary cases, α∈[2,4]; $\varepsilon_{W}$ describes the static power and the wake-up timing and wake-up frequency will affect the static power consumption, so $\varepsilon_{W}$ is inextricably related to scheduling strategies. Equation (Equation 9) implies that the energy depletion of the perceptron during the working duration is determined by the distance at which an object appears. If a sensor is in sleep mode, then only the low-power timer and other necessary components remain active, viz.
(10)$$e_{S}=\varepsilon_{S};$$
$\varepsilon_{S}$ represents the static power. Since frequent wake-up and sleep can affect the static power, that means $\varepsilon_{S}$ is also a function of the scheduling strategy.

The total energy depletion for the sensing circuit is going to be the weighted sum of these two,
(11)$$e=e_{S}S+e_{W}W,$$
where *W* and *S* represent working duration and sleeping duration, respectively. Then the mathematical expectation is:(12)$$E\left[e\right]=e_{S}\frac{S}{S+W}+e_{W}\frac{W}{S+W}.$$
Assign (Equation 9) and (Equation 10) to (Equation 11), we have:(13)$$E\left[e\right]=\varepsilon_{S}\frac{S}{S+W}+\left(\gamma {\left(d-r_{min}\right)}^{\alpha }+\varepsilon_{W}\right)\frac{W}{S+W}.$$

### 3.5. Scheduling Model

Figure 2 shows a schematic diagram of a typical alternate sensing scheduling scheme. A sensor wakes up its sensing unit when transitioning to the working state and turns it off when the sleep interval comes, and the length of the sleep interval is dynamically adjusted according to the event occurrence of the previous interval. Despite the scheme’s prevalence, how to adjust the intervals properly is as yet an open issue.

Now, we establish a baseline scheduling strategy: fixed-time scheduling (denoted as FT for convenience). In FT, we set the working duration and the sleep duration in advance and they remain unchanged. Then, we introduce an adaptive scheduling strategy, in which we dynamically fine-tune the sleep duration as per the event statistics and user-specified parameters. The mathematical formula is:(14)$$S_{k}=\left\{\begin{array}{c} \begin{array}{cc} S_{k-1}+\delta S_{k-1},\; & Sgn(W_{k-1})=0\\ S_{k-1}-\theta S_{k-1},\; & Sgn(W_{k-1})=1 \end{array} \end{array}\right.,$$
where δ and θ denotes the step-size increasing/diminishing factor, 0≤δ≤1, 0≤β≤1. Sgn(·) is sign function:(15)$$Sgn\left(t\right)=\left\{\begin{array}{c} \begin{array}{cc} 1,\; & \mathrm{If}\;\mathrm{an}\;\mathrm{event}\;\mathrm{is}\;\mathrm{detected}\;\mathrm{during}\;\mathrm{time}\;t\\ 0,\; & \mathrm{Otherwise} \end{array} \end{array}\right..$$
δ and θ provide the ability to maintain a synergy between the event capture and energy expenditure. For instance, a larger δ renders more energy-saving but also increases the probability of missing an event.

### 3.6. Sensing Coverage

Practical applications require the monitoring of almost the whole area of the wireless sensor network. Achieving 1-coverage is a minimum requirement (as shown in Figure 3).

**Theorem** **1.**
*Suppose there are N sensors uniformly distributed in a WSN, the sensing radius of each sensor is r, $r\in [r_{min},r_{max}]$, then the overall detection accuracy is given by:*
(16)$$\tilde{\mu }=1-\exp\left\{-\frac{W}{S+W}\frac{N\pi r^{2}}{M_{X}M_{Y}}\right\}.$$


**Proof.** The probability of a random event occurring within the sensing radii of $N_{0}$ sensors is:
(17)$$\mathrm{Pr}{}_{0}=\frac{{\left(\frac{N\pi r^{2}}{M_{X}M_{Y}}\right)}^{N_{0}}}{N_{0}}\exp\left\{-\frac{N\pi r^{2}}{M_{X}M_{Y}}\right\}.$$The expected number of sensors within the scope of an event is:
(18)$$E\left[N_{0}\right]=\frac{N}{M_{X}M_{Y}}\pi r^{2}.$$The expected number of sensors that are within the scope of an event and are themselves in a working state is:
(19)$$E\left[N_{0}^{W}\right]=-\frac{W}{S+W}\frac{N}{M_{X}M_{Y}}\pi r^{2}.$$Therefore, the probability of a random event being detected by a sensor is:
(20)$$1-\exp\left\{-\frac{W}{S+W}\frac{N\pi r^{2}}{M_{X}M_{Y}}\right\}.$$The theorem holds. □

Theorem 1 reveals that the detection accuracy is a function of the average number of sensors in each cell area and sensing scheduling.

### 3.7. Problem Formulation

The design objective of our model is an optimization problem:(21)$$\min\;J=\min\;\left\{\frac{1}{2}\left(1-\mu \right)+\frac{1}{2}\sigma \left(\frac{E\left[e\right]}{1000}\right)\right\}.$$

We change the optimization goal from maximizing μ to minimizing 1−μ. σ(·) is the logistic Sigmoid function, $\frac{1}{1+\exp(-x)}$, which maps the entire number line into a small range between 0 and 1. The reason we divide $E\left[e\right]$ by 1000 is that we want to convert the value of $E\left[e\right]$ to the linear region of the Sigmoid function—as can be seen from Figure 4, the linear region of $\frac{1}{1+\exp(-x)}$ is only 0–1 mW, while that of $\frac{1}{1+\exp(-x/1000)}$ is 0–4000 mW wide, which can fully cover the maximum variation range of energy depletion.

We list all the equations that determine *J*:(22)$$\left\{\begin{array}{c} \begin{array}{c} S=\sum_{n}^{k=1}S_{k}\\ S_{k}=\left\{\begin{array}{c} \begin{array}{cc} S_{k-1}+\delta S_{k-1},\; & Sgn(W_{k-1})=0\\ S_{k-1}-\theta S_{k-1},\; & Sgn(W_{k-1})=1 \end{array} \end{array}\right.\\ W=\sum_{n}^{k=1}W_{k}\\ W_{k}=\bar{w}\\ \mu =\left\{\begin{array}{c} \begin{array}{cc} \frac{W}{S+W}, & \mathrm{if}\;d<r_{\min}\\ \frac{W}{S+W}\exp\left\{-\xi_{1}{\left(d-r_{min}\right)}^{\xi_{2}}\right\}, & \mathrm{if}\;r_{\min}<d<r_{\max}\\ 0, & \mathrm{if}\;d>r_{\max} \end{array} \end{array}\right.\\ E\left[e\right]=\varepsilon_{S}\frac{S}{S+W}+\left(\gamma {\left(d-r_{min}\right)}^{\alpha }+\varepsilon_{W}\right)\frac{W}{S+W}\\ d\leq M_{X}\\ d\leq M_{Y}\\ 2\leq \alpha \leq 4\\ 0\leq \delta \leq 1\\ 0\leq \theta \leq 1 \end{array} \end{array}\right.$$

Since the occurrence of events is random, then:(23)$$\mathrm{Pr}\left\{Sgn\left(W_{k-1}\right)=0\right\}=\mathrm{Pr}\left\{Sgn\left(W_{k-1}\right)=1\right\}=\frac{1}{2},$$
which means
(24)$$\begin{array}{cc} S & =\sum_{k=1}^{n}S_{k}\\ \; & =\sum S_{k}\left|\left\{Sgn(W_{k-1})=0\right\}+\sum S_{k}\right|\left\{Sgn(W_{k-1})=1\right\}\\ \; & =\left(1+\frac{\delta -\theta }{2}\right)nS_{0}. \end{array}$$

The working interval is fixed to $\bar{w}$, so
(25)$$W=\sum_{k=1}^{n}W_{k}=n\bar{w}.$$

Put these together, then we get the final objective function:
(26)$$J=\left\{\begin{array}{c} \begin{array}{cc} \frac{1}{2}\frac{\left(1+\frac{\delta -\theta }{2}\right)nS_{0}}{\left(1+\frac{\delta -\theta }{2}\right)nS_{0}+n\bar{w}}+\frac{1}{2}\sigma \left(\frac{1}{1000}\frac{\left(\gamma {\left(d-r_{min}\right)}^{\alpha }+\varepsilon_{W}\right)n\bar{w}+\varepsilon_{S}\left(1+\frac{\delta -\theta }{2}\right)nS_{0}}{\left(1+\frac{\delta -\theta }{2}\right)nS_{0}+n\bar{w}}\right), & \mathrm{if}\;d<r_{\min}\\ \frac{1}{2}\left(1-\frac{n\bar{w}\exp\left\{-\xi_{1}{\left(d-r_{min}\right)}^{\xi_{2}}\right\}}{\left(1+\frac{\delta -\theta }{2}\right)nS_{0}+n\bar{w}}\right)+\frac{1}{2}\sigma \left(\frac{1}{1000}\frac{\left(\gamma {\left(d-r_{min}\right)}^{\alpha }+\varepsilon_{W}\right)n\bar{w}+\varepsilon_{S}\left(1+\frac{\delta -\theta }{2}\right)nS_{0}}{\left(1+\frac{\delta -\theta }{2}\right)nS_{0}+n\bar{w}}\right), & \mathrm{if}\;r_{\min}<d<r_{\max}\\ \frac{1}{2}\sigma \left(\frac{1}{1000}\frac{\left(\gamma {\left(d-r_{min}\right)}^{\alpha }+\varepsilon_{W}\right)n\bar{w}+\varepsilon_{S}\left(1+\frac{\delta -\theta }{2}\right)nS_{0}}{\left(1+\frac{\delta -\theta }{2}\right)nS_{0}+n\bar{w}}\right) & \mathrm{if}\;d>r_{\max} \end{array} \end{array}\right..$$

The constraint conditions are
(27)$$\begin{array}{cc} s.t.\; & 0\leq \delta \leq 1,\\ \; & 0\leq \theta \leq 1 \end{array}.$$
$\varepsilon_{W}$ and $\varepsilon_{S}$ are affected by the scheduling strategy. The question now is how to establish the functional relationship between $\varepsilon_{W}$, $\varepsilon_{S}$ and δ, θ. We have been thinking about numerical fitting methods—if we have real historical sample sensory readings, we can fit out the function between $\varepsilon_{W}$, $\varepsilon_{S}$ and δ, θ.

Numerical fitting is also known as curve fitting. As for the discrete sensory data collected by sampling, we often want to get a continuous function (i.e., a curve) or a denser discrete equation consistent with the known data. The steps of numerical fitting are:Presume a functional form. Commonly used function forms include: polynomial function, exponential function, logarithmic function, trigonometric function, and so forth.Determine the indeterminate coefficients. Using the least square method, point group center method, random fuzzy method, and so on, to determine the coefficients of the function. Of those, the least square method is the most commonly used.Evaluation. The imitative effect can be measured by the mean square error (MSE) or the degree of fit (R2).

## 4. Fibonacci Tree Optimization Algorithm

Recall that one-dimensional Fibonacci (golden ratio) search cannot proficiently take care of multi-variate issues, giving rise to Fibonacci Tree Optimization—an improved multi-dimensional search algorithm [44].

Let $X_{A}$, $X_{B}$ and $X_{C}$ be the vectors in D-dimensional Euclidean space. In particular, D=2. $X_{A}$ and $X_{B}$ address the endpoints of the search component fulfilling the optimization rule, and $X_{C}$ signifies the split points that can be resolved from searching rules. A proportion of the vectors can be determined as follows:(28)$$\frac{∥X_{C}-X_{A}∥}{∥X_{B}-X_{A}∥}=\frac{∥X_{B}-X_{C}∥}{∥X_{C}-X_{A}∥}=\frac{F_{p}}{F_{p+1}},$$
where $F_{p}$ denotes the *p*th Fibonacci number, and $\frac{F_{p}}{F_{p+1}}$ is equal to what is called the golden ratio.

The fitness function determined by the endpoints in the construction ought to be assessed as:(29)$$J\left(X_{A}\right)<J\left(X_{B}\right).$$
J(·) denotes the objective function. Then, the coordinate for split point $X_{C}$ can be calculated as:(30)$$X_{C}=X_{A}+\frac{F_{p}}{F_{p+1}}\left(X_{B}-X_{A}\right).$$

Finding the optimal value can likewise be viewed as setting up a search component in FTO, and can be partitioned into two phases: the local optimization phase and the global optimization phase. Allow Θ to signify the points set of the objective function and set ${\left|\Theta \right|}_{num}=F_{p},i=1,2,\cdots ,n$, where ${\left|\cdot \right|}_{num}$ denotes the number of points in the set, and *d* represents the depth of the tree. We select the point with the best fitness value. Then, in the following optimization stage, the points are rearranged by their fitness values from best to worst.

There are two search rules in FTO, which can be summed up:

**Rule One**: Consider the endpoints $X_{A}$ and $X_{B}$, which are given by
(31)$$\left\{X_{A}\right\}=\Theta =\left\{X_{k}|k\in \left[1,F_{p}\right]\right\}$$
(32)$$\left\{X_{B}\right\}=\Theta =\left\{X|X\in \prod_{j=1}^{D}\left[X_{lb}^{J},X_{ub}^{J}\right]\right\},$$
where Θ denotes the points set, with each point represented as $X_{k}$, of the *p*th iteration, and *k* is the index. $X_{A}$ absorb every point from Θ. $X_{B}$ select $F_{p}$ points from Θ randomly, where $F_{p}$ denotes the population size. $X_{ub}^{J}$ and $X_{lb}^{J}$ are the upper and lower limits for every point. For $\forall X\in \left\{X_{B}\right\}$, X satisfies a uniform distribution over the span $\left[X_{lb}^{J},X_{ub}^{J}\right]$, viz.
(33)$$\mathrm{Pr}\left(X\right)=U\left(X_{lb},X_{ub}\right)=\frac{1}{X_{ub}-X_{lb}}.$$

Using the $X_{A}$ and $X_{B}$ given above, the split points $X_{S1}$ are solved by Equation (Equation 30).

**Rule Two**: Assume that $X_{\mathrm{best}}$ is the optimal point in the current iteration given by rule one, viz.
(34)$$X_{\mathrm{best}}=BEST\left(\Theta \right).$$

Then, let $X_{A}=X_{\mathrm{best}}$; we have
(35)$$J\left\{X_{A}\right\}=\min\;\left\{J\left\{X_{k}\right\}|k\in \left[1,F_{p}\right]\right\}$$
(36)$$X_{B}=\left\{X_{k}|X_{k}\in \Theta \wedge X_{k}\neq X_{A}\right\}.$$

Hence, the split points $X_{S2}$ in the local optimization stage can be determined by Equations (Equation 35) and (Equation 36).

After applying the two rules above, new endpoints $X_{A}$ and $X_{B}$ and split points $X_{S1}$ and $X_{S2}$ are generated, and now we have $3F_{p}$ points. These points are sorted from best to worst based on the fitness value, retaining the optimal $F_{p}+1$ points, while the remaining $3F_{p}-F_{p}-1$ points are eliminated. At the end of this process, the set of search spaces for the current *p* iteration is updated from the remaining points and form a new set for the next iteration.

Figure 5 shows the branch generation process in the Fibonacci tree optimization algorithm. The depth is initialized as expected, and the number of points in each branch layer is equal to $F_{p}$. The dotted circle represents the search points set of the previous iteration; the solid red circle represents the endpoint $X_{A}$ of the current iteration, and the solid blue circles represent the global random endpoints. Figure 5a shows the global optimization phase, the split points $X_{S1}$ are constructed based on $X_{B}$ and $X_{A}$ and are represented by a solid white circle. Figure 5b shows the local optimization phase. New split points $X_{S2}$ can be obtained as per the rules.

The pseudo-code for the Fibonacci branch generation process is shown as Algorithm 1.
**Algorithm 1:** Pseudo-code of Fibonacci branch generation process
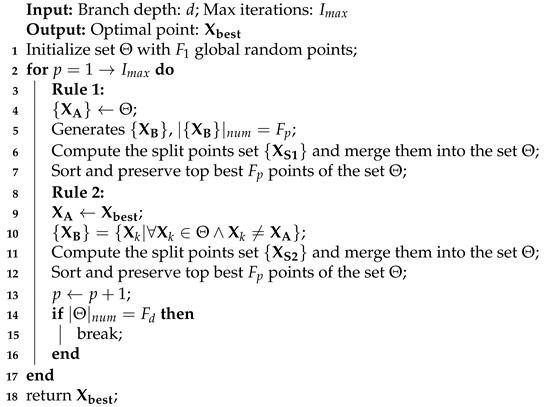


## 5. Analysis

### 5.1. Why FTO Algorithm Is Chosen for Parameter Optimization?

In the FTO algorithm, the growth of a Fibonacci tree is based on the optimal points of the latest generation, which is a process of competitive elimination. If there are peaks with different heights, the smaller peaks will be removed in the optimization process. Hence, the global optima can be achieved.The inner structure of the FTO algorithm meets the golden ratio all over. With each iteration, the separation of the golden ratio rapidly compresses the search space, providing the local optimal solution. It is adapted to the optimization of multi-modal functions. Conversely, most heuristic algorithms are based on the probabilistic search; they are essentially trial and error methods. It may take thousands or even millions of iterations, which is uneconomical and slow.The FTO setup only concerns the initial Fibonacci number, which is exquisite and simple. The typical way to obtain multiple local extremums of the multi-modal function using heuristic algorithms is the niche technology. Setting population parameters has a significant impact on the optimistic effects.Just like most heuristic algorithms, FTO is also inspired by natural phenomena—plant phototropism. Plants flourish more in sunny areas. To solve the optimization of the multi-modal function, FTO considers the global search space as a sunlit area. The hit probability is sustained and stable. The lush areas refer to the local search process, reflecting the fact that FTO can alternate between global and local searches.It fully utilizes the computer memory to save the optimization process and therefore is traceable.

The comparison between FTO and other optimization algorithms is listed in Table 5. Figure 6a illustrates the idea of golden separation in the FTO algorithm’s structure. Figure 6b reflects the global/local alternate search process inspired by phototropism of plants.

### 5.2. Why Do We Claim FTOS Strategy Is Superior to Other Scheduling Strategies?

Some of the latest scheduling strategies, such as LEACH, work under strict assumptions; for example, the network structure is flat, nodes are densely distributed, time is synchronized, and so forth. The FTOS strategy does not require time synchronization (centralized scheduling); each node independently determines the duty cycle; node density is not important; the location and distance information is not needed.The FTOS strategy determines the scheduling parameters as per the occurrence of events from the previous period, which is very scalable.Once the parameters are determined, they remain unchanged in the coming hours; you can easily deploy them.

Table 6 enlists the above-mentioned points.

## 6. Experiment

### 6.1. Experimental Setup

We set up a wireless sensor network comprising three nodes (N=3), covering a 5 m × 5 m lawn area (see Figure 7). Each node owns a single-chip microcomputer with a STM32F103ZET6 processor and a diffuse reflection photoelectric infrared proximity sensor. The technical parameters of the sensor are presented in Table 7.

We randomly throw objects into the area to simulate the event. The diffuse reflection photoelectric infrared proximity sensor with a sensing radius of *r* = 3~5 m ($r_{min}=3,r_{max}=5$) opens the switch when there is an object to block; the processor then issues a command to pull the buzzer.

Since the working voltage of the single-chip microcomputer is fixed to 5 V, the power formula is P=UI, so that the current reflects the size of the power consumption. Consequently, the energy is measured by the reading of the ammeter placed on each microcontroller.

### 6.2. Train

#### 6.2.1. Function Fitting

We assume that the sensor node determines its sleep interval based on the adaptive scheduling model mentioned in Section 3, where the parameters δ and θ are randomly generated on [0,1]. The polynomial function, double-exponential function, and checkmark function are used to fit the data (the forms are shown in Table 8). The determination of the coefficients is performed by the nonlinear least square method.

We generated 100 samples, recorded the static power losses, $\varepsilon_{S}$ and $\varepsilon_{W}$, and the corresponding parameters δ and θ. Figure 8 shows the fitting curves. Table 9 enlists the degree of fit, $R^{2}$.

Now that we have determined the functional relations between $\varepsilon_{S}$, $\varepsilon_{W}$ and δ, θ, the explicit form of the objective function *J* can be obtained by putting the fitted functions in.

#### 6.2.2. Univariate Analysis

Three sets of parameters, (0.175, 0.175), (0.65, 0.65), (0.5, 0.5), were used to test the sensing power dissipation against α in base sleep duration $\bar{s}$ and base working duration $\bar{w}$. The result is shown in Figure 9. When the distance between the event and the sensor, and the base sleep duration and base working duration, are fixed (d=3.15 m, $\bar{w}=1$ s, $\bar{s}=1$ s), the working power increases exponentially with α, and the sleeping power dissipation remains unchanged for α. With the increase of step-size parameter δ and θ, both the working power dissipation $e_{\bar{w}}$ and the sleeping power dissipation $e_{\bar{s}}$ decrease.

Figure 10 shows the variation of detection accuracy and expected energy depletion with δ. Figure 11 shows the variation of detection accuracy and expected energy depletion with θ. It can be seen that the two parts of the objective function *J*, 1−μ grow with δ, while $\sigma \left(E[e]/1000\right)$ decreases with δ; the optima can only be obtained by jointly optimizing 1−μ and $\sigma \left(E[e]/1000\right)$. 1−μ and $\sigma \left(E[e]/1000\right)$ move in the opposite direction against θ and $\bar{w}$ and $\bar{s}$ impose opposite effects on 1−μ and $\sigma \left(E[e]/1000\right)$. Anything that can increase/decrease 1−μ/$\sigma \left(E[e]/1000\right)$ by increasing $\bar{s}$ can also be achieved by decreasing $\bar{w}$.

Next, we set up a benchmark strategy: a fixed sleep time strategy, in which $S_{k}=\bar{s}=W_{k}=\bar{w}=1$ s. The relationship between the number of nodes *N*, the sensing radius *r* and the overall detection accuracy is studied. The results are shown in Figure 12. For both the proposed adaptive scheduling and fixed scheduling strategies, the detection accuracy will improve with the increase of the sensing radius and the number of nodes. However, the detection accuracy improvement of the proposed adaptive scheduling strategy is significantly higher than that of the fixed scheduling strategy.

#### 6.2.3. Optimization

By constructing a 3-D surface from the scattered data (see Figure 13), we found that J(δ,θ) is a bivariate multi-modal function, which favors the FTO algorithm.

Figure 14 shows the optimization process of the FTO algorithm, and the values of the horizontal and vertical axis represent the values of δ and θ, respectively. The darker the color of the contour map, the smaller the value of *J*. The algorithm stops after 500 iterations, and Figure 15 provides the final comparison of the peak areas. We can see that the peak area in the lower-left corner is the largest, and is where the optimal point is most likely to appear. The central point of the aforementioned area was taken as the ultimate global optimization point with the coordinate pair (δ,θ)=(0.175,0.175).

As a comparison, we also implemented the particle swarm optimization (PSO) algorithm, genetic algorithm (GA), comprehensive learning PSO (CLPSO) algorithm, differential evolution (DE) algorithm, and the artificial bee colony (ABC) algorithm. Table 10 outlines the parameters, and Table 11 presents their optimization results. The FTO algorithm outperforms others in most iteration rounds.

### 6.3. Test

Once the parameters of the strategy are determined, they remain unchanged in the coming hours.

Figure 16a presents the optimal *J* for the first four runs, applying ABC, DE, GA, PSO, CLPSO and FTO; Figure 16b shows a statistical result from 100 runs. From the perspective of distribution, FTO demonstrates a better optimistic effect than the other algorithms for its lowest center *J* value 0.47.

We have implemented several sensing scheduling strategies including LDAS, PECAS, BS and DSS. These strategies all make similar assumptions. Figure 17 shows the results. FTOS (δ=θ=0.175) accomplished better detection accuracy and consumes the bare minimum of energy compared to the other sensing scheduling strategies. For sensing duration (t = 20~150 s), all sensing scheduling strategies improved, viz., 1−μ and E[e] decline with time.

## 7. Deficiencies and Unresolved Issues

The research of this paper is still flawed in the following aspects:Selection of fitting functions. Some simple fitting functions are selected by experience to simulate the relationship between variables related to the objective function and the parameters of the scheduling strategy. The results are indeed good, but other fitting functions are still worth trying.Selection of sensors. The diffuse reflection photoelectric infrared proximity sensor, E3F-DS100P1, has a sensing radius of 3~5 m. If you have the budget to pay, you had better use sensors with a larger sensing radius and more accuracy, for example, laser radar.Too few nodes. Because of the insufficient number of devices we have, we have only configured a 3-node network. Experiments with more nodes would be more convincing.

The dynamics of network traffic were not analyzed. It is also interesting to associate our strategy with other approaches, such as the follow-up of specific targets in [19], which is also a topic for future efforts.

## 8. Conclusions

The FTOS strategy proposed in this paper addresses the synergy between the energy availability and fault tolerance of event monitoring. The experimental results show that the FTOS strategy contributes to reducing energy depletion and increasing detection accuracy. It outperforms LDAS, BS, DSS and PECAS, which have the same design objectives. The experimental findings show that the step-size parameters optimized by the FTO algorithm are better than those optimized by PSO, GA, and so forth. We also guide the actual deployment. It should be noted that when applied in practice, the FTOS strategy can be generalized only if the step-size can be systematically formulated.

## Figures and Tables

**Figure 1 sensors-21-05002-f001:**
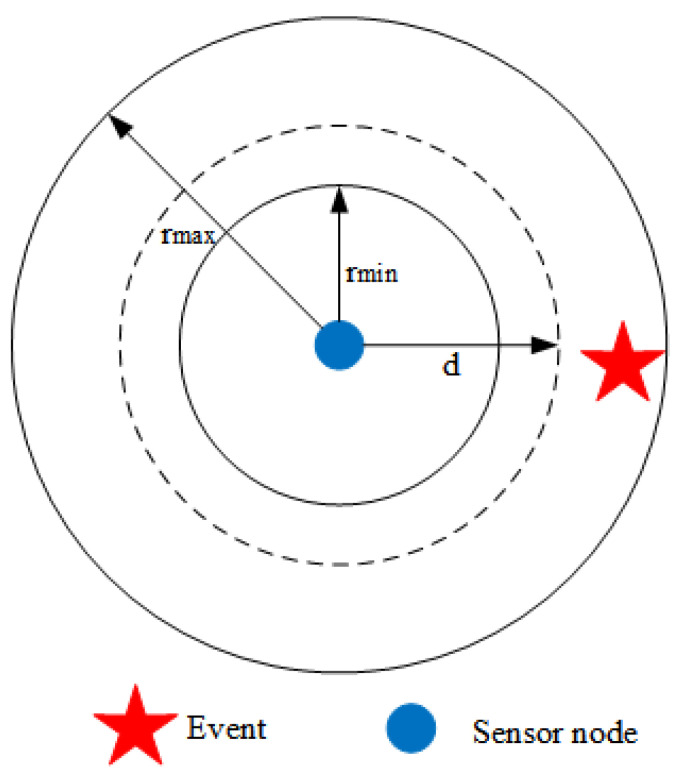
Tunable sensing radii.

**Figure 2 sensors-21-05002-f002:**
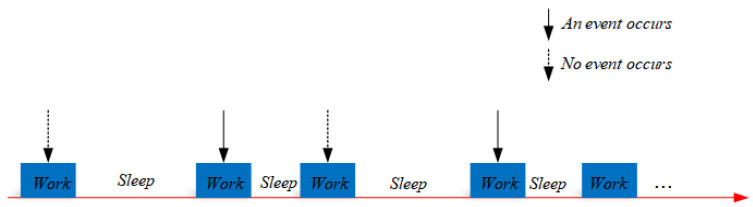
The diagram of alternate sensing scheduling.

**Figure 3 sensors-21-05002-f003:**
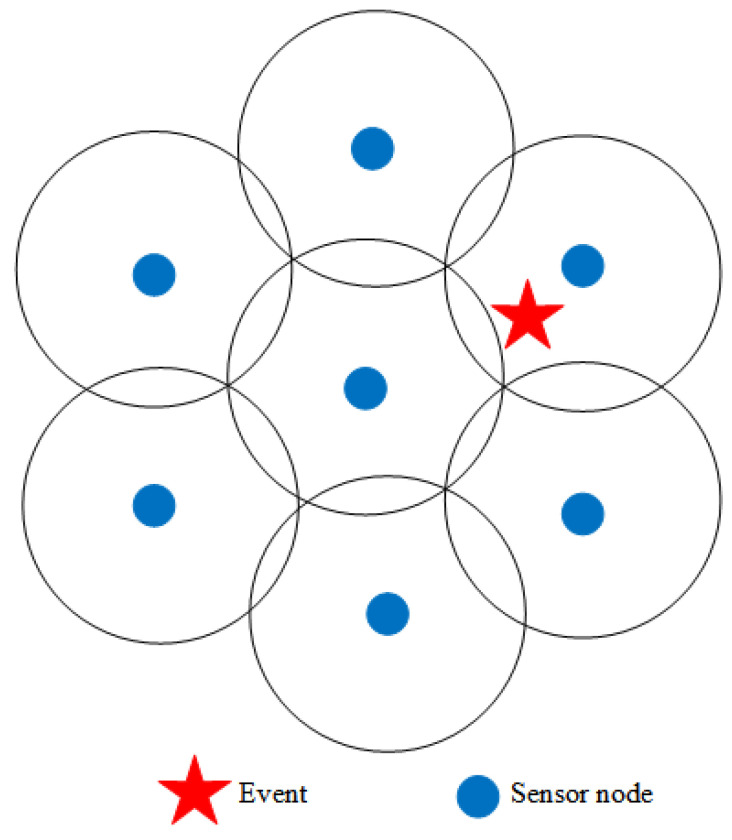
The diagram of 1-coverage.

**Figure 4 sensors-21-05002-f004:**
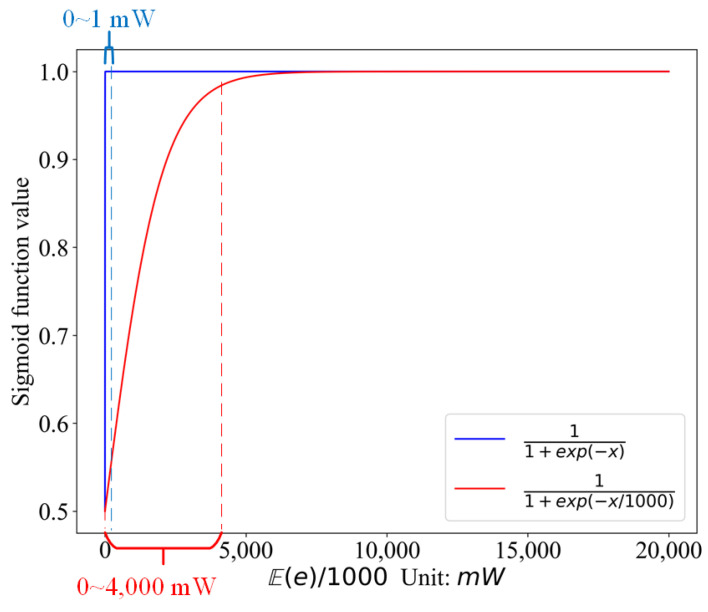
The liner region for different Sigmoid functions.

**Figure 5 sensors-21-05002-f005:**
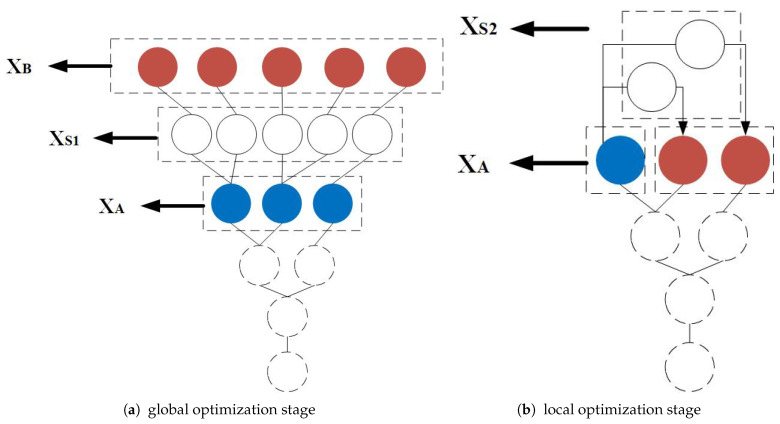
The diagram of the branch generation process in the Fibonacci tree optimization algorithm.

**Figure 6 sensors-21-05002-f006:**
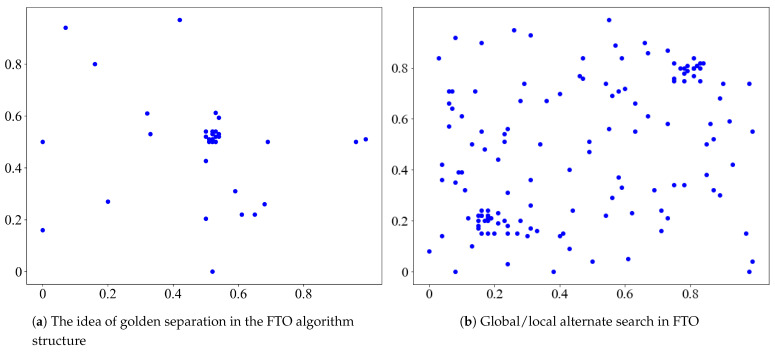
Optimal points in each iteration of FTO for optimizing: (**a**) $f\left(x,y\right)={\left(\frac{3}{0.05+9x^{2}+9y^{2}}\right)}^{2}+{\left(9x^{2}+9y^{2}\right)}^{2}$, and (**b**) $f\left(x,y\right)=0.5\left(\sin(10\ln(x))+\sin(10\ln(y))\right),x,y\in \left[0,1\right]$.

**Figure 7 sensors-21-05002-f007:**
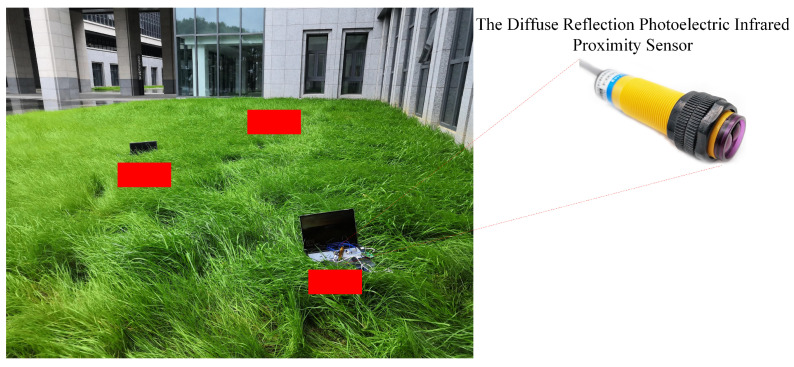
The experimental scene.

**Figure 8 sensors-21-05002-f008:**
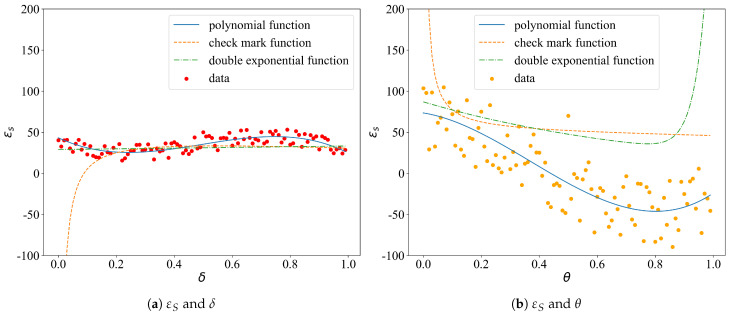
The fitting curves of $\varepsilon_{S}$, $\varepsilon_{W}$ and δ, θ.

**Figure 9 sensors-21-05002-f009:**
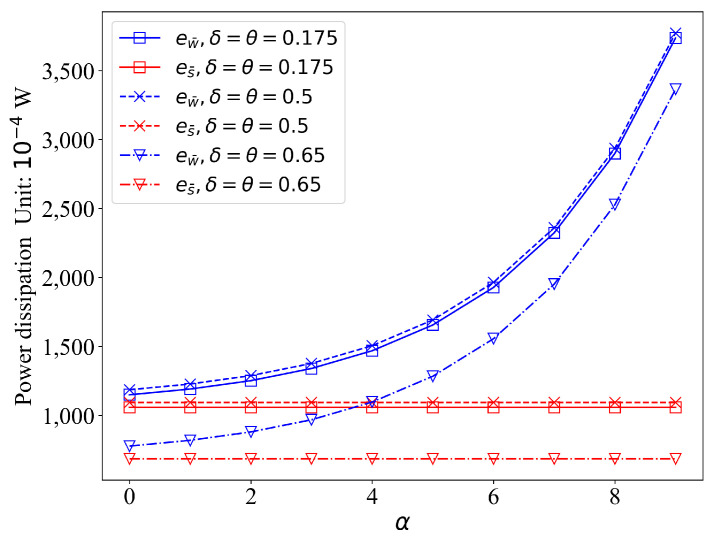
The power dissipation against α in base duration $\bar{s}$ and $\bar{w}$ (d=3.15 m, $\bar{w}=1$ s, $\bar{s}=1$ s).

**Figure 10 sensors-21-05002-f010:**
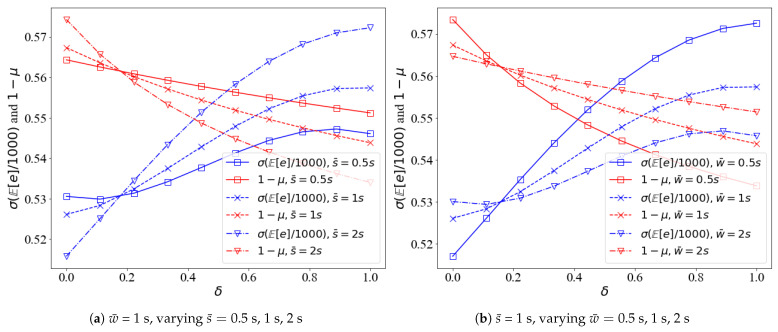
The detection accuracy and energy depletion against δ. (**a**) $\bar{w}$ is fixed to 1 s, varying $\bar{s}$; (**b**) $\bar{s}$ is fixed to 1 s, varying $\bar{w}$.

**Figure 11 sensors-21-05002-f011:**
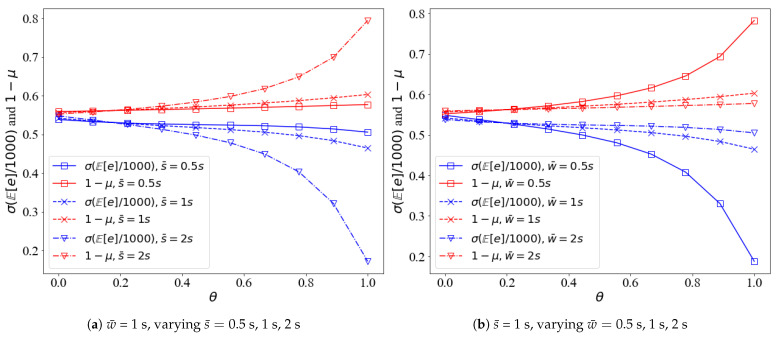
The detection accuracy and energy depletion against θ. (**a**) $\bar{w}$ is fixed to 1 s, varying $\bar{s}$; (**b**) $\bar{s}$ is fixed to 1 s, varying $\bar{w}$.

**Figure 12 sensors-21-05002-f012:**
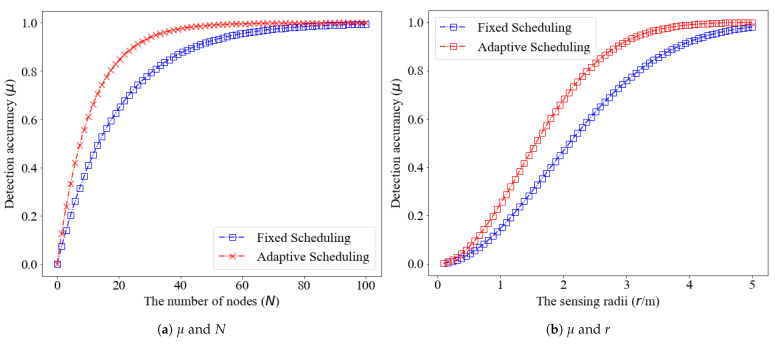
Detection accuracy for the proposed adaptive scheduling and fixed scheduling (**a**) against the number of cognitive sensors *N* (*r* is fixed and r=3.15 m); and (**b**) sensing radius *r* (*N* is fixed and N=100).

**Figure 13 sensors-21-05002-f013:**
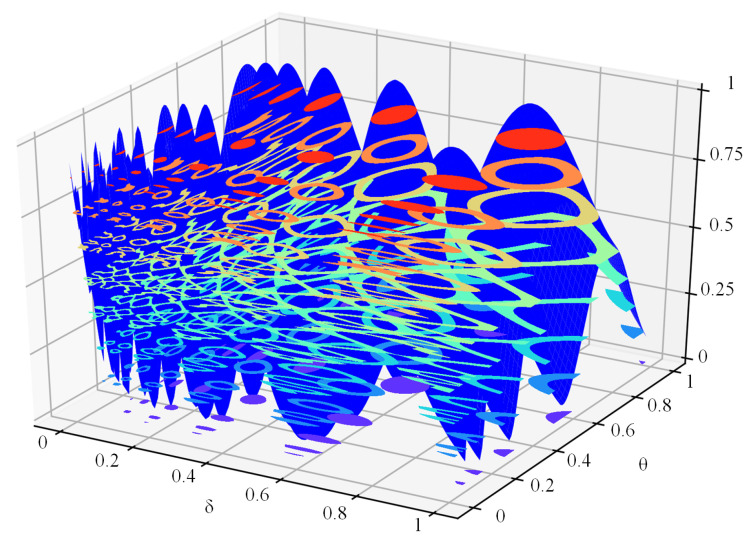
3-D surface graph for scattered data.

**Figure 14 sensors-21-05002-f014:**
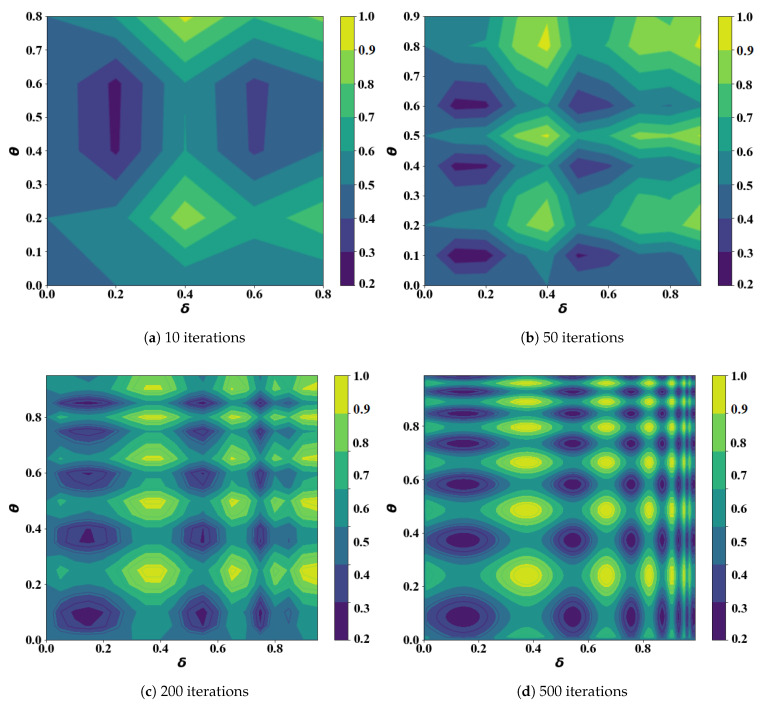
The contour plots in iterations.

**Figure 15 sensors-21-05002-f015:**
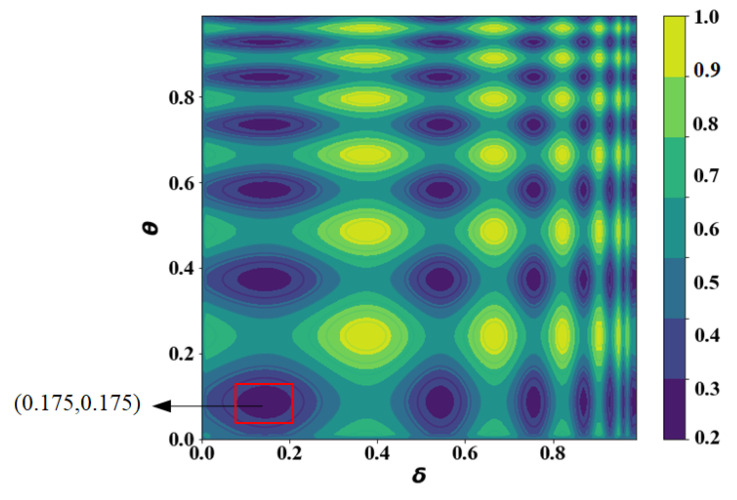
The determination of the ultimate optimum point.

**Figure 16 sensors-21-05002-f016:**
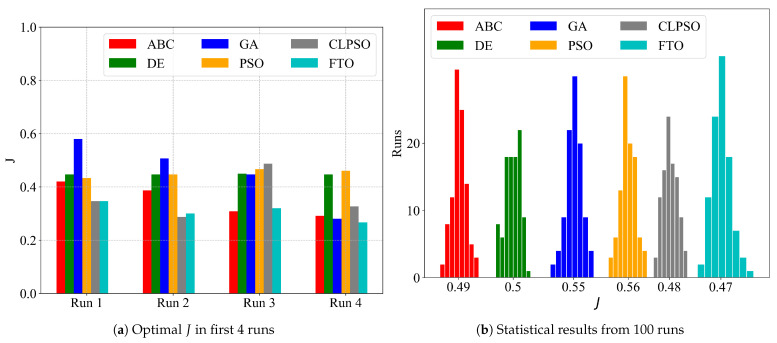
Test results of FTOS scheduling with parameters optimized by different algorithms.

**Figure 17 sensors-21-05002-f017:**
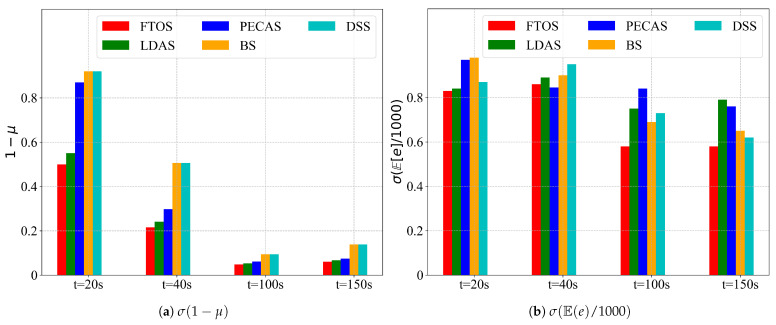
The expected detection accuracy and the expected energy depletion of different sensing scheduling strategies at different time scales.

**Table 1 sensors-21-05002-t001:** The studies’ classifications according to their assumptions.

Strategies	Assumptions								
	Network Structure	Sensor Placement	High Density	Sensing Area	Time Synch	Frequent Failures	Mobility	Known Location	Known Distance
RIS [18]	Flat	Grid, Uniform, Poisson	N	2-D	Y	N	N	N	N
BS [20]	Hierarchical	Poisson	N	2-D	Y	N	N	N	Y
LDAS [15]	Flat	Uniform	N	2-D	N	N	N	N	N
PEAS [16]	Flat	Uniform	Y	Any	N	Y	N	N	N
PECAS [17]	Flat	Uniform	N	Any	N	Y	N	N	N
CCP [21]	Flat	Any	N	2-D	N	N	N	Y	N
ASCENT [18]	Flat	Any	Y	Any	N	N	N	N	N
CSS [22]	Flat	Any	Y	2-D	Y	N	N	Y	N
LEACH-GA [23]	Hierarchical	Any	N	Any	Y	N	N	N	N
IBLEACH [24]	Hierarchical	Any	N	Any	Y	N	N	N	N
ESS [25]	Hierarchical	Any	N	Any	Y	N	N	N	N
DSS [26]	Hierarchical	Poisson	N	2-D	Y	N	N	N	Y
IDCA [27]	Flat	Any	N	2-D	N	N	N	Y	N

Y: yes; N: no.

**Table 2 sensors-21-05002-t002:** The studies’ classifications according to their objectives.

Strategies	Objectives				
	Sensing Coverage	Network Connectivity	Simplicity	Robustness	Energy Balance
RIS [18]	*k*-coverage,asymptotic	×	×	Y	Y
BS [20]	×	×	×	×	Y
LDAS [15]	1-coverage	×	×	×	Y
PEAS [16]	×	1	×	×	×
PECAS [17]	×	1	×	×	Y
CCP [21]	*k*-coverage	*k*	×	×	Y
ASCENT [18]	×	×	Y	×	×
CSS [22]	*k*-coverage	*k*	×	Y	Y
LEACH-GA [23]	×	×	×	Y	Y
IBLEACH [24]	×	*k*	×	Y	Y
ESS [25]	×	×	×	Y	Y
DSS [26]	×	×	×	×	Y
IDCA [27]	Full-coverage	×	×	×	Y

×: not mentioned.

**Table 3 sensors-21-05002-t003:** Comparison of some optimization algorithms.

Algorithm	Type	Searching Philosophy	Global Search Ability	Local Search Ability	Incorporate Niching Techniques	Convergence Speed
GA [33]	evolutionary intelligence	probabilistic search	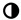	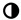	No	Slow
DE [34]	evolutionary intelligence	probabilistic search	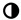	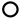	No	Slow
PSO [35]	swarm intelligence	probabilistic search	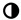	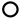	No	Slow
CLPSO [36]	swarm intelligence	probabilistic search	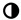	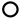	No	Slow
ABC [37]	swarm intelligence	probabilistic search	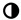	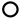	No	Slow
MLGSA [38]	swarm intelligence	probabilistic search	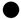	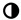	No	Fast
ACS-NP [41]	swarm intelligence	probabilistic search	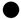	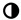	Yes	Fast
CLPSO-LS [42]	swarm intelligence	probabilistic search	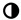	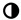	Yes	Fast
QTS [39]	evolutionary intelligence	probabilistic search	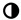	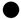	No	Fast
FTO [44]	Recursive search	recursive search	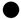	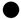	No	Fast

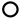
: weak; 
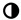
: medium; 
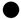
: strong.

**Table 4 sensors-21-05002-t004:** List of essential notations in the model descriptions.

#	Description
*W*	Total working duration
*S*	Total sleep duration
$W_{k}$	The *k*-th working interval
$S_{k}$	The *k*-th sleep interval
*n*	Number of intervals
$S_{0}$	Initial sleep interval (fixed)
*s*	Average sleep duration per duty cycle
*w*	Average working duration per duty cycle
*M*	The interested 2-D square region
$M_{x}$	Length of *M*
$M_{y}$	Width of *M*
C(t)	The counting process of events in [0,t]
$\hat{q}{|}_{\left[0,t\right]}$	Expected number of events processed in [0,t]
$\Gamma_{s}$	Average number of missed events in sleep duration
$\Gamma_{w}$	Average number of missed events in working duration
Γ	Event space
λ	Poisson rate
*d*	Distance between sensor and event
*r*	Sensing radius
$r_{min}$	Minimum sensing radius
$r_{max}$	Maximum sensing radius
μ	Detection accuracy
$\xi_{1},\xi_{2}$	Metrics of detection ability
*e*	Energy depletion for sensing
$e_{S}$	Energy depletion of sensing in sleep duration
$e_{W}$	Energy depletion of sensing in working duration
$\varepsilon_{S}$	Static power loss in sleep mode
$\varepsilon_{W}$	Static power loss in working mode
γ	Coefficient of energy depletion growing with distance
α	Changeable power scaling parameter for the sensing circuit
δ	Step-size increasing factor
θ	Step-size diminishing factor
*N*	Number of sensors
Sgn(·)	The sign function
σ(·)	The logistic Sigmoid function

**Table 5 sensors-21-05002-t005:** The comparison between FTO and other optimization algorithms.

	FTO	Heuristic Algorithms
Parameters	simple	many, hard to determine
Imitation of nature	phototropism	swarm intelligence
Characteristic	competitive elimination alternate search golden ratio search	random search based on probability
Global search capability	strong	prone to be trapped in local optimum
Convergence speed	fast	slow
Resources footprint	low	high

**Table 6 sensors-21-05002-t006:** The differences between FTOS and other scheduling strategies.

		FTOS	Other Scheduling Strategies
Assumptions	Time synchronization	no	yes
	Node density	no requirement	high
	Transmission radii	tunable	untunable
	Node location	movable	immovable
Adaptability		strong	weak
Deployment		easy	difficult

**Table 7 sensors-21-05002-t007:** The technical parameters for the diffuse reflection photoelectric infrared proximity sensor.

Parameter	Value
Product model	E3F-DS100P1
Operating voltage	DC 6~36 V
Operating current	200 mA
Response frequency	50 Hz
Sensing object	Any opaque object
Sensing radius	10~100 cm, adjustable

**Table 8 sensors-21-05002-t008:** The fitting functions.

Function	Form
Polynomial function	$ax^{5}+bx^{4}+cx^{3}+dx^{2}+ex+f$
Double-exponential function	$ae^{bx}+ce^{dx}$
Checkmark function	$ax+\frac{b}{x}+c$

**Table 9 sensors-21-05002-t009:** The degree of fit ($R^{2}$).

Cases	Polynomial Function	Double-Exponential Function	Checkmark Function
$\varepsilon_{S}$ and δ	0.87	0.93	0.35
$\varepsilon_{S}$ and θ	0.83	0.80	0.56
$\varepsilon_{W}$ and δ	0.73	0.32	0.46
$\varepsilon_{W}$ and θ	0.54	0.76	0.93

**Table 10 sensors-21-05002-t010:** The algorithm parameters adopted.

Algorithm	Parameter	Value
Particle Swarm Optimization (PSO)	Population	20
	Cognitive ratio	2
	Social coefficient	2
	Inertia weight	0.4~0.9
Genetic Algorithm (GA)	Population	100
	Point mutation	0.01
	Hoist mutation	0.4
	Parsimony coefficient	0.01
	Learning rate	0.05~0.5
Comprehensive Learning Particle Swarm Optimization (CLPSO)	Population	20
	Cognitive ratio	2
	Social coefficient	2
	Inertia weight	0.4~0.9
Differential Evolution (DE)	Population	20
	Scaling factor	0.6
	Crossover rate	0.8
Artificial Bee Colony (ABC)	Followers	50
	Scouters	20
	Employed foragers	1
Fibonacci Tree Optimization (FTO)	Nested branch depth	2
	Total branch depth	6
	Search space	[0,1]
	Max iterations	1000
	Precision	0.001

**Table 11 sensors-21-05002-t011:** The optimization results.

Algorithm	Iter = 10			Iter = 50			Iter = 200			Iter = 500		
	min J	δ	θ	min J	δ	θ	min J	δ	θ	min J	δ	θ
PSO	0.65	0.57	0.165	0.67	0.71	0.11	0.55	0.75	0.34	0.56	0.59	0.65
GA	0.87	0.57	0.2	0.76	0.38	0.6	0.57	0.56	0.13	0.55	0.55	0.65
CLPSO	0.52	0.72	0.4	0.49	0.18	0.74	0.56	0.75	0.9	0.54	0.15	0.15
DE	0.67	0.57	0.37	0.67	0.78	0.56	0.5	0.44	0.15	055	0.65	0.65
ABC	0.63	0.57	0.38	0.58	0.51	0.17	0.49	0.45	0.65	0.49	0.65	0.5
FTO	0.52	0.74	0.56	0.45	0.18	0.86	0.45	0.8	0.13	0.45	0.175	0.175

## Data Availability

Research data are be obatained by sending an email to wuliangshun@whu.edu.cn.

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
