# Peer review of "Energy-Efficient Adaptive Sensing Scheduling in Wireless Sensor Networks Using Fibonacci Tree Optimization Algorithm"

_sensors, 2021, doi:10.3390/s21155002_

Round 1
Reviewer 1 Report
This paper studies energy-efficient adaptive sensing scheduling in Wireless Sensor Networks (WSN). The Fibonacci tree optimization algorithm is utilized, and it is compared with other heuristic algorithms. Overall, it is well-organized research. Some detailed comments to further improve it are as follows.
- Line 11, it is better to use “multimodal” than “multi-peak”.
- Line 14, please provide the full name of FTOS.
- Line 25, please provide the full name of MAC.
- Lines 46-51, it is recommended to separate each section to be a sentence.
- Lines 56-57 should be two sentences. “The duty cycle includes active time, nap time, and idle listening period of the detectors; the goal is to best model the event arrival patterns.” Please try to avoid using “;”.
- Line 106, please provide the full name of LDAS, PEAS, and PECAS.
- Lines 132-134, please provide the full name of all abbreviations. Any references for BS?
- P5, Section 2.2, there are many heuristic algorithms. It is helpful to extend the review for existing algorithms. Some references are recommended to be included. (1) (1) Comprehensive learning particle swarm optimization algorithm with local search for multimodal functions. IEEE Transactions on Evolutionary Computation 23 (4), 718-731, 2019. (2) A multi-layered gravitational search algorithm for function optimization and real-world problems. IEEE/CAA Journal of Automatica Sinica, 8(1), 94-109, 2020, etc. Thus, please enhance the reason for utilizing the Fibonacci tree optimization algorithm.
- Please revise lines 201-202 by not using “;”.
- In Figure 1, what does the difference in the vertical direction mean?
- Line 215, the “box symbol” after the sentence should be removed.
- P16, please add citations for the compared algorithms.
- How many independent runs have been carried out for each algorithm? This should be clearly provided. A statistical result from multiple runs should be presented.
Author Response
Dear reviewer,
We appreciate your pertinent advices, and all of your comments have been incorporated into this manuscript.
Attched please find our point-by-point response to your comments.
Best regards,
H. J. Cai

Reviewer 2 Report
The Abstract needs a restructure. It is not easy to find out what is the main contribution of the paper.
Please briefly mention other optimization approaches, particularly nature-based ones and highlight the advantage of Fibonacci Tree Optimization Algorithm over other optimization approaches such as Ant colony (An improved model of ant colony optimization using a novel pheromone update strategy, IEICE), or Neural networks as explained in (Socialization of industrial robots: An innovative solution to improve productivity)
Figures are undesirable too large. For example, fig 1 is too large and low resolution.
Please explain 6 in more detail.
Author Response
Dear reviewer,
We appreciate your pertinent advices, and all of your comments have been incorporated into this manuscript.
Please see the attachment for our point-by-point response to your comments.
Best regards,
H. J. Cai

Reviewer 3 Report
I suggest rewriting the article since there are notable coincidences with other articles in the same field.
Round 2
Reviewer 3 Report
The presented manuscript applies Fibonacci tree optimization algorithm to minimizing average energy expenditure and maximizing sensing coverage. Later, algorithm is compared with CLPSO, DE, LDAS, BS, DSS and PECAS.
The main conclusion is that the trade-off between energy availability and fault tolerance of event monitoring is better than the strategies above-mentioned.
The article has been substantially improved both in the expressions and in some confusing parts of the text.
The J-cost function has been adequately reformulated and differs from that applied in other works.
I think it is missing some formal issues that should be addressed:
1- Review the figures numbers and its caption.
2- Review the tables.
3- The supplementary material is not well upload.
